# Possible Neuroprotective Effects of l-Carnitine on White-Matter Microstructural Damage and Cognitive Decline in Hemodialysis Patients

**DOI:** 10.3390/nu13041292

**Published:** 2021-04-14

**Authors:** Yuji Ueno, Asami Saito, Junichiro Nakata, Koji Kamagata, Daisuke Taniguchi, Yumiko Motoi, Hiroaki Io, Christina Andica, Atsuhiko Shindo, Kenta Shiina, Nobukazu Miyamoto, Kazuo Yamashiro, Takao Urabe, Yusuke Suzuki, Shigeki Aoki, Nobutaka Hattori

**Affiliations:** 1Department of Neurology, Juntendo University Faculty of Medicine, Tokyo 113-8421, Japan; dtanigu@juntendo.ac.jp (D.T.); motoi@juntendo.ac.jp (Y.M.); at-shindo@juntendo.ac.jp (A.S.); k-shiina@juntendo.ac.jp (K.S.); nobu-m@juntendo.ac.jp (N.M.); nhattori@juntendo.ac.jp (N.H.); 2Department of Radiology, Juntendo University Faculty of Medicine, Tokyo 113-8421, Japan; nqg16274@nifty.com (A.S.); kkamagat@juntendo.ac.jp (K.K.); andicach@gmail.com (C.A.); saoki@juntendo.ac.jp (S.A.); 3Department of Neurology and Stroke Medicine, Graduate School of Medicine, Yokohama City University, Yokohama 236-0004, Japan; 4Department of Nephrology, Juntendo University Faculty of Medicine, Tokyo 113-8421, Japan; jnakata@juntendo.ac.jp (J.N.); yusuke@juntendo.ac.jp (Y.S.); 5Department of Nephrology, Juntendo University Nerima Hospital, Tokyo 177-8521, Japan; hiroaki@juntendo.ac.jp; 6Department of Neurology, Juntendo University Urayasu Hospital, Urayasu 279-0021, Japan; kazuo-y@juntendo.ac.jp (K.Y.); t_urabe@juntendo.ac.jp (T.U.)

**Keywords:** l-carnitine, hemodialysis, vascular dementia, diffusion tensor imaging, diffusion kurtosis imaging, neurite orientation dispersion and density imaging

## Abstract

Although l-carnitine alleviated white-matter lesions in an experimental study, the treatment effects of l-carnitine on white-matter microstructural damage and cognitive decline in hemodialysis patients are unknown. Using novel diffusion magnetic resonance imaging (dMRI) techniques, white-matter microstructural changes together with cognitive decline in hemodialysis patients and the effects of l-carnitine on such disorders were investigated. Fourteen hemodialysis patients underwent dMRI and laboratory and neuropsychological tests, which were compared across seven patients each in two groups according to duration of l-carnitine treatment: (1) no or short-term l-carnitine treatment (NSTLC), and (2) long-term l-carnitine treatment (LTLC). Ten age- and sex-matched controls were enrolled. Compared to controls, microstructural disorders of white matter were widely detected on dMRI of patients. An autopsy study of one patient in the NSTLC group showed rarefaction of myelinated fibers in white matter. With LTLC, microstructural damage on dMRI was alleviated along with lower levels of high-sensitivity C-reactive protein and substantial increases in carnitine levels. The LTLC group showed better achievement on trail making test A, which was correlated with amelioration of disorders in some white-matter tracts. Novel dMRI tractography detected abnormalities of white-matter tracts after hemodialysis. Long-term treatment with l-carnitine might alleviate white-matter microstructural damage and cognitive impairment in hemodialysis patients.

## 1. Introduction

Correlated with the substantial increase in patients with end-stage renal disease due to diabetes mellitus, hypertension, and obesity, the number of patients on maintenance dialysis has risen greatly, to 284 individuals per million population worldwide [1]. Cognitive impairment is critical in hemodialysis patients and is associated with death and dialysis withdrawal [2]. A previous large-scale cohort study of hemodialysis patients documented the prevalence of diagnosed dementia as 4%, whereas several studies investigated cognitive function tests of hemodialysis patients, showing cognitive impairment in as many as 87% [2,3,4]. Thus, there may be more undiagnosed or covert vascular dementia in hemodialysis patients than expected.

Diffusion tensor imaging (DTI) and diffusion kurtosis imaging (DKI) allow noninvasive investigation of the neural architecture of the brain using a Gaussian and non-Gaussian model, respectively [5,6]. Furthermore, multi-shell diffusion MRI techniques (msdMRI), including the neurite orientation dispersion and density imaging (NODDI) model consisting of intracellular volume fraction (ICVF), orientation dispersion index (ODI), and isotropic volume fraction (ISO), have been shown to more sensitively evaluate neuritic microstructure alterations than DTI [7]. Using DTI, the association of white-matter fiber tract disorders with cognitive decline has been reported in vascular dementia, as well as small vessel diseases in hemodialysis patients [8,9,10,11], although no evidence is currently available for the analysis of white-matter fiber tracts using msdMRI techniques.

In experimental studies, we recently created rat models of vascular dementia by ligation of bilateral common carotid arteries, creating chronic cerebral hypoperfusion, which induced cerebral white-matter damage and vascular dementia [12,13]. l-Carnitine was shown to play a pivotal role in suppressing inflammation, oxidative stress, and apoptosis in chronic debilitating diseases [14]. l-Carnitine also improves cardiac dysfunction, as well as peripheral artery disease [15,16]. Importantly, continuous supplementation with l-carnitine for a longer period alleviated oxidative stress in oligodendrocytes and promoted axonal outgrowth and myelination in cerebral white matter, thereby improving cognitive impairment after chronic cerebral hypoperfusion in rats [13]. So far, no specific therapeutic agents for vascular cognitive dysfunction associated with cerebral white-matter lesions are available in clinical practice.

Cognitive dysfunction in hemodialysis patients can be induced by a disorder of cerebral circulation due to atherosclerosis in cerebral arteries, oxidative stress, inflammation, and possibly blood pressure fluctuation due to hemodialysis, which might share common pathological mechanisms with chronic cerebral hypoperfusion in our experimental studies [12,13]. Taking advantage of the utility of novel dMRI techniques, we sought to assess the alterations of the microstructure of white-matter tracts in hemodialysis patients. Importantly, a postmortem brain study of a hemodialysis patient enrolled in the present study was carried out. Meanwhile, several studies with small sample sizes of 10 to 17 participants were conducted to explore the pathophysiology of microstructural damage of white-matter tracts using NODDI [17,18]. l-Carnitine alleviated white-matter lesions and cognitive impairment in an experimental study, but its effects on cognitive decline in hemodialysis patients are essentially unknown. The aim of the current study was to elucidate the therapeutic efficacy of l-carnitine for disorders of white-matter tracts seen on dMRI and cognitive decline in hemodialysis patients.

## 2. Materials and Methods

### 2.1. Study Participants

Patients were included on the basis of the following inclusion criteria, (i) age between 55 and 90 years, (ii) undergoing hemodialysis in Juntendo University Hospital Dialysis Center; and (iii) having at least one atherosclerotic vascular risk factor. Exclusion criteria were as follows: (i) any contraindication to brain MRI, (ii) moderate to severe cognitive decline or aphasia (need some assistance to complete their daily living activities or communication), (iii) coexistence of neurological diseases, and (iv) presence of severe pneumonia, chronic heart failure, or advanced cancer. Previous studies with sample sizes of 10 to 17 participants were conducted to explore the pathophysiology of microstructural alterations of white-matter tracts using NODDI [17,18]. Thus, 14 maintenance hemodialysis patients (age 70.8 ± 7.2 years, six women and eight men), treated three times a week at outpatient hemodialysis units in Juntendo University Hospital, were included in the present study. In addition, 10 age- and sex-matched healthy controls (73.0 ± 4.4 years, five women and five men) who were literate or able to be interviewed and could communicate effectively, with no history of severe mental disorders or dementia, were also recruited. This study was conducted in accordance with the Declaration of Helsinki. The independent ethics committee of Juntendo University Hospital approved this study (16-170). All study subjects were given an explanation of the study, and written, informed consent was obtained for the study objective, MRI and cognitive function tests, enrolment in the study, and ensuring confidentiality of information.

### 2.2. Risk Factors

Atherosclerotic vascular risk factors were defined according to the previous literature [19]: (1) hypertension, history of using antihypertensive agents, systolic blood pressure >140 mmHg, or diastolic blood pressure >90 mmHg; (2) diabetes mellitus, use of oral hypoglycemic agents or insulin, or glycosylated hemoglobin >6.4%; (3) dyslipidemia, use of antihyperlipidemic agents, serum LDL-C ≥140 mg/dL, HDL-C <40 mg/dL, or triglycerides ≥150 mg/dL; (4) current smoking status; (5) coronary artery disease, previous history of angina pectoris or myocardial infarction; (6) stroke, previous history of ischemic stroke, hemorrhagic stroke, or undetermined stroke.

### 2.3. Drug Administration and Classification

The protocol for treatment with l-carnitine (Otsuka Pharmaceutical Co., Ltd., Tokushima, Japan) was 600 mg orally once a day starting in July 2012, followed by 1000 mg intravenously per hemodialysis day starting in January 2014. Termination of l-carnitine treatment was determined randomly, and patients without l-carnitine treatment were also included. On the basis of the duration of l-carnitine treatment, seven patients each were classified into the no or short-term l-carnitine treatment (NSTLC) and LTLC long-term l-carnitine treatment (LTLC) groups, with an allocation ratio of 1:1. In the NSTLC group, four patients’ intravenous l-carnitine treatment was stopped from June 2014 to April 2015; thus, l-carnitine was discontinued on the day of dMRI and neuropsychological tests (January 2017 to September 2017), whereas the remaining three patients were not treated with l-carnitine. In the LTLC group, intravenous l-carnitine treatment was continued on the day of dMRI and neuropsychological tests (January 2017 to September 2017) in seven patients.

### 2.4. Profile of Carnitine Kinetics after l-Carnitine Treatment and Other Laboratory Blood Examinations

Serum total, free, and acylated carnitine concentrations were measured using the enzyme cycling method, and the ratio of acylated carnitine to free carnitine was calculated. According to the above treatment protocol of l-carnitine, total, free, and acylated carnitine concentrations were measured before treatment, at 6 months after 600 mg of l-carnitine treatment orally per day, and at 6 months after 1000 mg of l-carnitine treatment intravenously per hemodialysis day. Other laboratory data (calcium, phosphate, glucose, low-density lipoprotein, high-density lipoprotein, hemoglobin, triglycerides, creatinine, eGFR, leukocyte count, and high-sensitivity C-reactive protein) were examined by standard enzymatic methods. The averages of these data in three consecutive tests before MRI were calculated.

### 2.5. Neuropsychological Tests

The mini mental state examination (MMSE), Hasegawa dementia rating scale-revised (HDS-R) for standard cognitive function, the frontal assessment battery (FAB), and the Japanese version of the Montreal cognitive assessment (MoCA-J) were performed. The trail making test (TMT) was also conducted, consisting of two parts. TMT part A is a task of rapid visual scanning and processing speed, in which the respondent connects randomly arranged numbers in consecutive order. In part B, the respondent connects randomly arranged letters and numbers in consecutive order, alternating between numbers and letters. The four subtests of the Wechsler Memory Scale-Revised (WMS-R), namely, logical memory I/II, digit span, and visual span, were also carried out. All neuropsychological tests were performed by an experienced neuropsychologist (MS) within 60 min for each patient. The standard scores of TMT and four subsets of WMS-R were calculated by average score and standard deviation for each age group from previous data of Japanese subjects (Appendix A) [20], using the formula Z (standard score) = *x* (raw score) − μ (mean score)/σ (standard deviation).

### 2.6. MRI Acquisition

MRI data were acquired on a 3 T MRI scanner (MAGNETOM Prisma; Siemens Healthcare, Erlangen, Germany) with a 64-channel head coil. Sequences of conventional MRI included fluid-attenuated inversion recovery imaging (FLAIR) and the GRE T2 * sequence. FLAIR (TR/TE = 10,000/114 ms) was used to evaluate the degree of deep and subcortical white-matter hyperintensity (DSWMH) and periventricular hyperintensity (PVH) in accordance with Fazekas grades 0–3 [21], and GRE T2 * (TR/TE = 410–500/12 ms) was used to identify cerebral microbleeds (CMBs), defined as rounded areas of signal loss with diameter <10 mm. Diffusion-weighted imaging (DWI) was performed by spin-echo echo-planar imaging in the anterior-to-posterior phase-encoding direction. Multi-shell DWI was obtained with b-values of 1000 and 2000 s/mm^2^ along 64 isotropic diffusion gradients for each shell. Each DWI acquisition was completed with a gradient-free image (b = 0). The acquisition parameters were as follows: repetition time/echo time = 3300/70 ms/ms, field of view = 229 × 229 mm, matrix = 130 × 130, resolution = 1.8 × 1.8 mm, slice thickness = 1.8 mm, and acquisition time = 7.29 min.

### 2.7. MRI Preprocessing

The diffusion datasets were corrected for eddy-current distortions and small head movements and denoised using EDDY and TOPUP toolboxes [22]. The resulting images were visually checked in axial, coronal, and sagittal views. All images were free of severe artefacts, such as gross geometric distortion, signal dropout, or bulk motion. The resulting images were used to acquire (1) fractional anisotropy (FA), MD (mean diffusivity), AD (axial diffusivity), and RD (radial diffusivity) maps using ordinary least squares applied to the diffusion-weighted images with b = 0 and 1000 s/mm^2^, (2) mean kurtosis (MK), axial kurtosis (AK), and radial kurtosis (RK) using Diffusion Kurtosis Estimator (https://www.nitrc.org/projects/dke/, accessed on 1 June 2018), and (3) ICVF, ODI, and ISO maps using the NODDI model implemented in NODDI Matlab Toolbox5 (http://www.nitrc.org/projects/noddi_toolbox, accessed on 1 April 2019) and AMICO.

### 2.8. Tract-Based Spatial Statistics Analysis

Tract-based spatial statistics (TBSS), a voxel-wise statistical analysis of whole-brain white matter, was implemented in FMRIB Software Library version 5.0.9 (FSL; Oxford Centre for Functional MRI of the Brain, Oxford, UK; www.fmrib.ox.ac.uk/fsl, accessed on 1 September 2019) [21]. TBSS was performed to identify significant differences between groups of each of the DTI, DKI, and NODDI indices.

### 2.9. Tract-of-Interest (TOI) Analysis

Any maps showing significant clusters by TBSS analysis were localized using the Johns Hopkins University’s ICBM-DTI-81 WM tractography atlas, which is composed of the left (right) anterior thalamic radiation (L[R]atr), corticospinal tract (L[R]cs), cingulum cingulate gyrus (L[R]cc), cingulum hippocampus (L[R]ch), forceps major (fm), forceps minor (fmi), inferior fronto-occipital fasciculus (L[R]ifof), inferior longitudinal fasciculus (L[R]ilf), superior longitudinal fasciculus (L[R]slf), uncinate fasciculus (L[R]uf), and slf temporal part (L[R]slftemp). The diffusion measures were averaged in each TOI delineated by the atlas. Correlation analysis was performed to explore the relationships between each index and neuropsychological scores [23].

### 2.10. Histopathological Analysis of an Autopsy Brain


An autopsy brain from a patient was fixed with 15% neutral buffered formalin, and the selected sections were embedded in paraffin. The paraffin-embedded blocks were sliced at a thickness of 6 μm. Brain sections were stained with hematoxylin and eosin (H&E), Klüber–Barrera (KB), and immunohistochemical stains using several antibodies for specific proteins. For immunohistochemistry, brain sections underwent antigen retrieval by heat activation in an autoclave with or without formic acid pretreatment, before being incubated overnight at 4 °C in primary antibody. The primary antibodies used were against phosphorylated tau, amyloid beta, and neurofilament heavy weight (NFH). Bound antibodies were visualized using the peroxidase-polymer-based method using a Histofine Simple Stain MAX-PO kit (Nichirei, Tokyo, Japan) with diaminobenzidine as the chromogen.

### 2.11. Statistical Analysis

Values presented in this study are expressed as means ± standard deviation. All statistical analyses were performed using IBM SPSS Statistics for Windows, version 22.0 (IBM Corporation, Armonk, NY, USA), except for general linear model (GLM) analysis, which was performed using the FSL. The Shapiro–Wilk test was used to assess data normality, whereas demographic data were analyzed using the unpaired Student’s *t*-test for continuous variables, and the chi-squared test was used for categorical variables. The effect size was then calculated using Cohen’s *d* to evaluate the statistical power of the relationship determined during group comparisons [24]. Clinical features, values on psychological tests, laboratory data, and degree of PVH and DSWMH were analyzed using the Mann–Whitney test for nonparametric variables and the chi-squared test for categorical variables between hemodialysis patients in the NSTLC and LTLC groups. For TBSS analysis, a GLM framework with one-way analysis of variance (ANOVA; healthy controls vs. hemodialysis patients with NSTLC vs. hemodialysis patients with LTLC) with age and sex as covariates were applied in the FSL randomize tool with 5000 permutations. The results were then corrected for multiple comparisons by controlling family-wise error and applying the threshold-free cluster enhancement option. For TOI analysis, one-way ANOVA with the Welch and Games–Howell post hoc tests was performed to compare among the three groups. Spearman’s rank correlation coefficient was used to test for any linear associations between the diffusion metrics and psychological test scores that reached significant differences between the NSTLC and LTLC groups. The false discovery rate (FDR) was used to correct for multiple comparisons (20 TOIs). In all analyses, a probability value <0.05 was considered significant.

## 3. Results

### 3.1. Study Population

The baseline characteristics of the 14 enrolled patients are summarized in Table 1; 93%, 43%, and 57% of the patients had hypertension, diabetes mellitus, and dyslipidemia, respectively, and 14% and 21% of patients had coronary artery and peripheral artery diseases, respectively. Causes of hemodialysis were diabetic nephropathy in six patients, chronic glomerulonephritis in three patients, and nephrosclerosis, polycystic kidney disease, rapidly progressive glomerulonephritis, and renal cell carcinoma in one patient each.

### 3.2. Profile of Serum Carnitine Kinetics before and after l-Carnitine Treatment and Classification

In 11 patients, treatment with 600 mg of l-carnitine orally per day was started in July 2012, followed by 1000 mg of l-carnitine intravenously per hemodialysis day from January 2014, whereas the remaining three patients were not treated with l-carnitine. The total amount of l-carnitine was significantly higher in the LTLC group than in the NSTLC group (751.4 ± 62.6 g vs. 241.3 ± 233.1 g, *p* = 0.002, Table 1). In the 11 patients who were treated with l-carnitine, serum levels of total carnitine were 47.4 ± 12.1 μmol/L at baseline, increased to 195.0 ± 57.2 μmol/L at 6 months after 600 mg of oral l-carnitine treatment (*p* < 0.01), and further increased substantially to 548.9 ± 148.2 μmol/L at 6 months after 1000 mg of intravenous l-carnitine treatment (*p* < 0.001, vs. baseline and after oral l-carnitine treatment, respectively, Figure 1). Free and acylated carnitine levels were 29.7 ± 10.3 μmol/L and 17.7 ± 3.1 μmol/L at baseline, increased to 123.8 ± 40.6 μmol/L and 71.2 ± 23.1 μmol/L after 600 mg of oral l-carnitine treatment (*p* < 0.01 and *p* < 0.05, respectively), and further increased to 332.2 ± 84.3 μmol/L and 216.8 ± 74.5 μmol/L after 1000 mg of intravenous l-carnitine treatment, respectively (*p* < 0.001, vs. baseline and after oral l-carnitine treatment, respectively, Figure 1). The ratio of acylated carnitine to free carnitine did not differ after l-carnitine treatment.

### 3.3. Baseline Patients’ Profile, Degree of White-Matter Lesions, Psychological Tests, and Laboratory Data in the NSTLC and LTLC Groups

Between the NSTLC and LTLC groups, age, frequency of male sex, final education level, atherosclerotic risk factors, vascular diseases, and cause of renal failure were not significantly different (Table 1). On laboratory data, high-sensitivity C-reactive protein (hs-CRP) levels were significantly lower in hemodialysis patients with LTLC than in those with NSTLC (0.17 ± 0.52 vs. 0.65 ± 0.68 mg/dL, *p* < 0.001), whereas other laboratory markers did not show any significant differences. Degrees of PVH and DSWMH and number of CMBs were not different between groups.

### 3.4. Cognitive Functions in Hemodialysis Patients with NSTLC and LTLC

Overall, MMSE, HDS-R, FAB, and MoCA-J scores were 25.4 ± 3.5, 26.3 ± 2.3, 14.9 ± 3.2, and 23.9 ± 3.2 points, respectively, which were not different between the NSTLC and LTLC groups (Table 1). In TMT-A, completion time was 76.2 ± 53.5 s and 50.4 ± 21.3 s in the NSTLC and LTLC groups, respectively. *Z*-Scores were significantly lower in the LTLC group than in the NSTLC group (−0.8 ± 1.2 vs. 2.2 ± 3.6, *p* = 0.017, Figure 2). On TMT-B, completion times were 138.9 ± 77.7 s and 159.8 ± 107.1 s in the LTLC and NSTLC groups, respectively, whereas one patient in the NSTLC group was unable to complete the TMT-B test. *Z*-Scores were not significantly different between the LTLC and NSTLC groups (0.3 ± 2.3 vs. 0.4 ± 1.4, Figure 2). In the four subtests of the WMS-R including logical memory I, logical memory II, digit span, and visual memory span, each score and the corresponding *Z*-score were 18.4 ± 7.5, −0.2 ± 1.0, 12.1 ± 3.3 and −0.1 ± 0.9, and 15.4 ± 2.2, 0 ± 0.7, 14.1 ± 7.7, and −0.1 ± 1.0 points, respectively, and there were no significant differences between the LTLC and NSTLC groups (Figure 2).

### 3.5. DTI Analysis in Hemodialysis Patients with NSTLC and LTLC

On TBSS, as shown in Figure 3, group comparison showed that there were lower FA and higher AD, RD, and MD in the NSTLC group than in the HC group in many white-matter tracts, whereas there were no significant differences between the LTLC and NSTLC groups and between the HC and LTLC groups (Table 2). TOI analyses showed that FA values in Latr, Ratr, Lcs, Rcs, Lcc, Rcc, fm, fmi, Lifof, Rifof, Lilf, Lslf, Rslf, and Luf were lower in the NSTLC group than in the HC group (*p* < 0.05, Appendix A). There were no differences in AD values (Appendix A). RD values in Latr, Ratr, Lcs, Rcs, Lcc, Rcc, fm, fmi, Lifof, Rifof, Lilf, Lslf, Rslf, Luf, and Lslftemp, and in fm and Rslf were higher in the NSTLC group and in both the NSTLC and LTLC groups than in the HC group, respectively (*p* < 0.05, Appendix A). MD values in Latr, Ratr, fm, fmi, Lifof, Rifof, Lilf, Lslf, Luf, Ruf, Lslftemp, and Rslftemp were higher in the NSTLC group, and MD values in Rslf were higher in the NSTLC and LTLC groups than in the HC group (*p* < 0.05, Appendix A).

### 3.6. DKI Analysis in Hemodialysis Patients with NSTLC and LTLC

On TBSS, AK was lower in the LTLC and NSTLC groups than in the HC group (Figure 4A,B, Table 2). In RK and MK, bilateral white-matter tracts were widely lower in the NSTLC group than in the HC group, whereas RK in a confined tract was lower in the LTLC group than in the HC group, and there was no difference in MK between the LTLC and HC groups (Figure 4C–E, Table 2). On TOI analyses, RK values in Lcc and Rifof and in Ratr, Rcc, and Lslf were lower in the NSTLC group and in both the NSTLC and LTLC groups than in the HC group (*p* < 0.05), respectively, whereas there were no differences in AK and MK values (Appendix A).

### 3.7. NODDI Analysis in Hemodialysis Patients with NSTLC and LTLC

On TBSS analysis, the NSTLC and LTLC groups showed significantly lower ICVF in bilateral subcortical white matter tracts than the HC group (Figure 5A,B, Table 2). In TOI analyses, ICVF values in Latr, Ratr, Lcs, Rcs, Lcc, Rcc, fm, fmi, Lifof, Rifof, Lilf, Rilf, Lslf, Rslf, Lslftemp, and Rslftemp were lower in both or either of the NSTLC and LTLC groups than in the HC group (*p* < 0.05, Appendix A). In ODI analyses, there were no significant differences among the three groups on TBSS and TOI analyses (Appendix A). In ISO analyses, the LTLC group showed lower values in right temporal, frontal, and occipital subcortical tracts than the HC and NSTLC groups (Figure 5C,D, Table 2), whereas there were no differences in TOI values (Appendix A).

### 3.8. Association of Cognitive Function Tests with dMRI in the Comparison between NSTLC and LTLC Groups

Among the types of cognitive function tests, the LTLC group showed better achievement of TMT-A than the NSTLC group. Significant correlations were shown for the associations of completion times of TMT-A with Ratr (*r* = 0.79, *p* = 0.001) in AD, Lcc (*r* = −0.75, *p* = 0.002), Latr (*r* = −0.77, *p* = 0.001), and Rslf (*r* = −0.76, *p* = 0.002) in AK, and fm (*r* = −0.79, *p* = 0.001) in ICVF (Table 3).

### 3.9. Histopathological Characteristics of an Autopsy Brain

A 75 years old Japanese man with NSTLC, with a previous history of diabetes mellitus, diabetic retinopathy, and nephropathy, developed intracranial hemorrhage in the right thalamus and was admitted to our hospital. Despite acute therapy for thalamic hemorrhage, such as intravenous blood pressure-lowering drugs, glycerin, and hemostatic agents, the thalamic hemorrhage worsened, and he died 9 days after admission. Postmortem examination showed that right thalamic hemorrhage was comorbid with intraventricular rupture, but cerebral white matter was preserved depending on the location where microscopic analyses were carried out (Figure 6A,B). On KB staining, moderate rarefaction of white matter together with loss of myelin sheaths was found in the subcortical white matter of the superior frontal gyrus (Figure 6C,D). The NFH staining showed that axons were less sparse in the superior frontal gyrus (Figure 6E). Phosphorylated tau inclusions and accumulations of amyloid beta in the hippocampus were not apparent (Figure 6F,G).

## 4. Discussion

### 4.1. Main Findings

The present study first explored the disorder of white-matter tracts using dMRI and the impact of l-carnitine treatment for the white-matter tract injuries, as well as cognitive dysfunction, in hemodialysis patients. The principal findings of the current study were that impairment of white-matter tracts was robustly detected in hemodialysis patients with pathological confirmation in an autopsy case, and these injuries were alleviated by treatment with LTLC, along with reduction of hs-CRP levels. Furthermore, hemodialysis patients treated with LTLC displayed better performance on TMT-A than hemodialysis patients with NSTLC, and specific white-matter tracts that contributed to the achievement of TMA-A were identified.

Previous studies showed that hemodialysis patients had mean MMSE scores ranging from 20.7 to 25.7 points, and 24–87% of hemodialysis populations were categorized as having dementia [2,3,4,25,26,27,28]. In the current study, MMSE and MoCA-J scores were 25.4 ± 3.5 and 23.9 ± 3.2 points, respectively, and the majority of the cases were considered to have mild cognitive impairment, because the cutoff values of MMSE and MoCA-J were found to be 27 and 25 points, respectively [29,30,31]. Importantly, the current data showed that low FA, high RD and MD, and low RK were clearly seen in hemodialysis patients, particularly in those treated with NSTLC. Furthermore, an autopsy study showed rarefaction of white matter and loss of myelin sheaths without the pathological hallmarks of Alzheimer’s disease, which were consistent with those of vascular dementia [12,13]. Thus, the present data showed that damage to microstructural organizations in cerebral white matter can be a fundamental cause of cognitive decline in hemodialysis patients, which was clearly shown by the correlation between dMRI and histopathology.

Several mechanisms for the development of cognitive dysfunction related to white-matter diseases in hemodialysis patients have been proposed. First, concomitant atherosclerotic vascular risk factors contributed to the cerebral small vessel diseases yielding silent infarcts, white-matter lesions, and microbleeds, thereby causing vascular dementia [32,33,34,35]. Second, hemodialysis itself altered cerebral blood flow, which was previously shown by positron emission tomography and transcranial Doppler [36,37]. Induction of acute fluid shifts, intravascular volume loss, and brain edema, thereby leading to disorders of cerebral tissue and metabolism, could occur. Third, oxidative stress, chronic inflammation, and coagulopathy that were increased in end-stage renal disease may induce endothelial injury and neuronal damage [3]. Alternatively, uremic toxins such as hyperhomocysteinemia, guanidine compound, and cystatin-C have a role in neurodegeneration [38]. The present data showed that long-term l-carnitine therapy improved white-matter structural damage and TMT-A performance, along with a reduction in serum hs-CRP levels. It was shown that l-carnitine reduced markers of oxidative stress and inflammation in end-stage renal disease [14]. Our previous experimental research demonstrated that continuous supplementation of l-carnitine for 28 days after cerebral chronic hypoperfusion resulted in suppression of oxidative stress in oligodendrocytes, facilitation of myelin sheaths and axonal outgrowth via the phosphatase and tensin homologue deleted on chromosome ten/Akt/mammalian target of rapamycin pathway, protection from mitochondrial dysfunction, and improvement of cognitive dysfunction. Generally, 80% of the total plasma carnitine is free carnitine, and 20% is acylated carnitine, with a normal acyl/free carnitine ratio of 0.25 [39]. Free carnitine not only transports long-chain fatty acids to mitochondria for subsequent β-oxidation, but also acts as a scavenger by eliminating toxic acyl compounds from mitochondria; thus, an acyl/free carnitine ratio >0.4 indicates carnitine insufficiency [39]. In the present data, although the acyl/free carnitine ratio was still >0.4 after l-carnitine therapy, substantial increases in free carnitine levels could robustly increase β-oxidation and ATP synthesis, thereby exerting pleiotropic effects and protecting white-matter microstructure in hemodialysis patients. Furthermore, the present data suggest that l-carnitine could be a potential candidate for novel therapeutic agents for vascular cognitive dysfunction other than in hemodialysis patients.

In the present comprehensive dMRI data, white-matter microstructural damage showing low FA, high RD and MD, and low RK on DTI and DKI was more critical in NSTLC. On the other hand, NODDI showed comparably low ICVF in LTLC and NSTLC, whereas relatively lower ISO in LTLC comparing with NSTLC was seen in limited tracts. ICVF indicates the density of neurites including axons and dendrites based on intracellular diffusion, and ISO indicates the volume fraction of isotropic diffusion [7]. Considering the previous experimental and the current results, although considerably lower neurite density due to damage of neural tissues could occur, long-term treatment with l-carnitine may alleviate injuries of white-matter microstructures, possibly due to suppression of neuroinflammation, in hemodialysis patients. However, the sample size in the current study was small, and a future prospective study with a larger sample size is warranted.

The LTLC group showed better achievement on the TMT-A than hemodialysis patients in the NSTLC groups. TMT is a task to evaluate the ability to flexibly switch attention between competing task-set representations and is denoted by the total time to completion [40]. The TMT-A is considered to reflect a baseline measure of psychomotor speed, visuospatial search, and target-directed motor tracking [40,41]. Fundamentally, multiple cognitive processes are thought to be implicated in performing the TMT; therefore, different brain regions and connecting fiber tracts are associated with completing the TMT. Lesion–symptom mapping studies showed that the right frontal area, left rostral anterior cingulate cortex, and left dorsomedial prefrontal lobe were related to impairments of TMT performance in patients with central nervous system disorders such as stroke and brain tumors [42,43,44]. The present comprehensive dMRI data showed that completion times for TMT-A correlated with disorders of bilateral anterior thalamic radiations, the left cingulum in the cingulate gyrus, the right superior longitudinal fasciculus, and forceps major. Thus, the combination of DTI, DKI, and NODDI elucidated the neuroanatomical connections related to performance on the TMT-A in hemodialysis patients for the first time.

### 4.2. Limitations

Some potential limitations of the current study must be considered when interpreting the present results. First, the sample size was small, and this study was a nonrandomized, cross-sectional study analyzing the therapeutic effects of l-carnitine for cognitive decline and microstructural damage on dMRI in hemodialysis patients. Although there was a significant difference in the total amount of l-carnitine between the LTLC and NSTLC groups, the protocol of l-carnitine treatment was quite complicated, and the NSTLC group was heterogenous in that it consisted of patients without l-carnitine treatment and patients with short-term l-carnitine treatment. Thus, to explore the protective effects of l-carnitine for disorders of white-matter fiber tracts and cognitive function, large-scale, prospective longitudinal studies with an appropriate trial design are warranted. Furthermore, it is important to explore the association of serum carnitine concentrations with microstructural damage of cerebral white matter. Second, only one postmortem examination was performed in the current study. The types of end-stage renal diseases in the enrolled patients varied; therefore, there may have been heterogeneity in postmortem brain pathology. Third, other medications including statins, antiplatelet drugs, and angiotensin II receptor blockers might have antiatherogenic effects and could have affected the microstructural damage of white-matter tracts and the neuropsychological status in hemodialysis patients.

## 5. Conclusions

The current data showed that l-carnitine may have neuroprotective roles against the white-matter microstructural damage and cognitive impairment in hemodialysis patients. It was found that DTI, DKI, and NODDI could certainly detect abnormalities of white-matter fiber tracts in hemodialysis patients, consistent with rarefaction of white matter and loss of myelin sheaths on histopathology. Long-term treatment with l-carnitine may ameliorate white-matter microstructural injuries by possibly suppressing neuroinflammation and improve achievement of TMT-A performance linked with protecting some candidate fiber tracts. Long-term treatment with l-carnitine can be a novel therapeutic approach for vascular cognitive dementia in hemodialysis patients.

## Figures and Tables

**Figure 1 nutrients-13-01292-f001:**
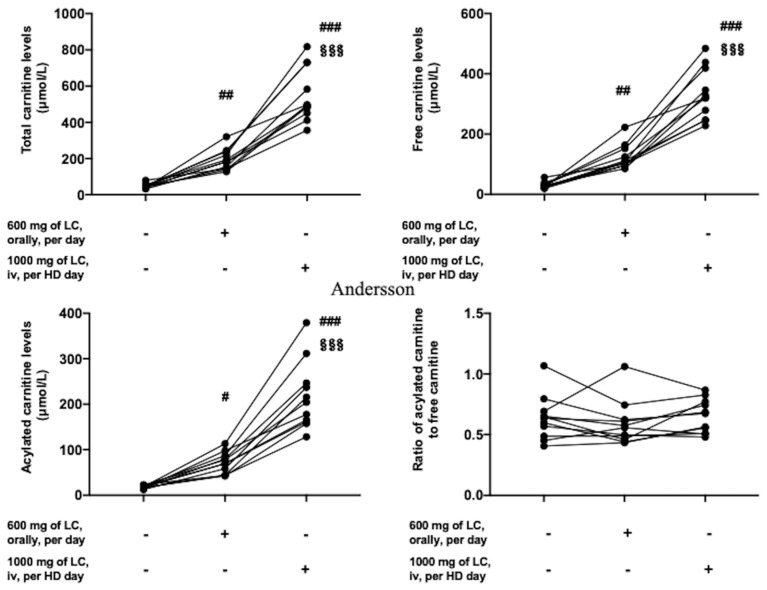
Temporal profile of carnitine kinetics after l-carnitine treatment. Serum total, free, and acylated carnitine concentrations were measured, and the ratios of acylated carnitine to free carnitine were calculated before l-carnitine treatment, at 6 months after 600 mg of l-carnitine treatment orally a day, and at 6 months after 1000 mg of l-carnitine treatment per hemodialysis day. # *p* < 0.05, ## *p* < 0.01, ### *p* < 0.001 vs. at baseline; §§§ *p* < 0.001, vs. at 6 months after 600 mg of l-carnitine treatment orally. LC = l-carnitine; HD = hemodialysis.

**Figure 2 nutrients-13-01292-f002:**
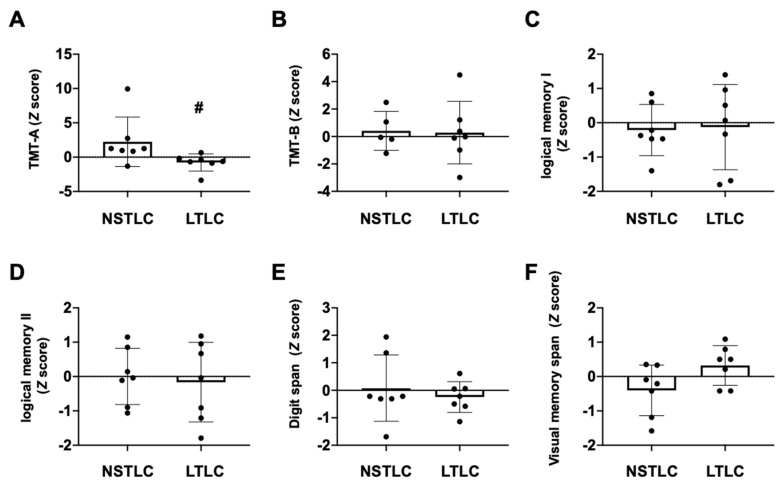
Comparison of neuropsychological tests between patients treated with NSTLC and with LTLC. Dot plots showing *Z* (standard score) of TMT-A (**A**), TMT-B (**B**), subsets of WMS-R logical memory I (**C**)/II (**D**), digit span (**E**), and visual span (**F**) between hemodialysis patients treated with NSTLC and with LTLC. The Mann–Whitney test was used for comparison. *Z* (Standard score) = *x* (raw score) − μ (mean score)/σ (standard deviation); # *p* < 0.05. LTLC, long-term l-carnitine treatment; NSTLC, no or short-term l-carnitine treatment; TMT = trail making test.

**Figure 3 nutrients-13-01292-f003:**
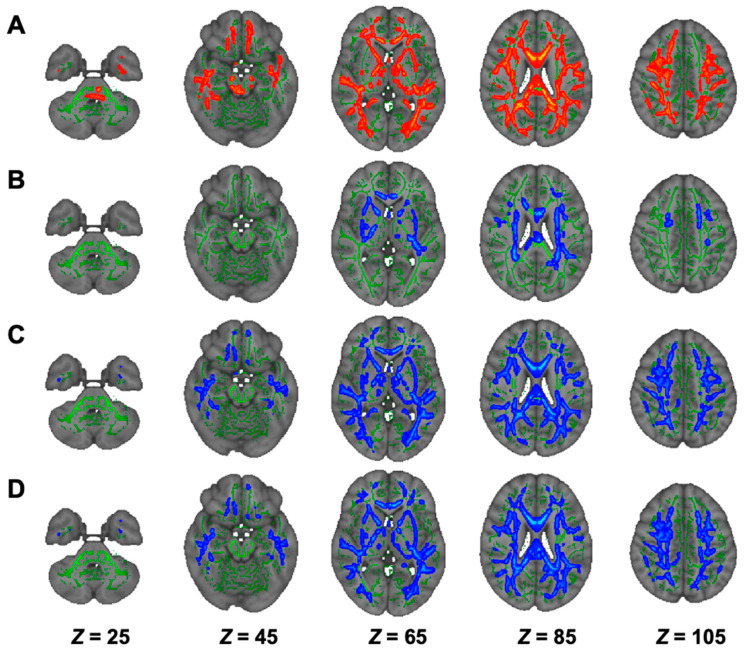
Comparison of TBSS of DTI among hemodialysis patients treated with NSTLC and with LTLC and healthy controls. Comparisons of DTI (FA, (**A**); AD, (**B**); RD, (**C**); and MD, (**D**)) indices among hemodialysis patients treated with NSTLC and LTLC and healthy controls are shown. TBSS analyses show that hemodialysis patients treated with NSTLC have significantly lower FA and significantly higher MD, AD, and RD than healthy controls (*p* < 0.05, (**A**–**D**)), whereas there are no significant differences between hemodialysis patients treated with NSTLC and LTLC, and hemodialysis patients treated with LTLC and healthy controls (data not shown). The skeleton is presented in green. To aid visualization, the results are thickened using the fill script implemented in the FMRIB Software Library. AD, axial diffusivity; DTI, diffusion tensor imaging; FA, fractional anisotropy; LTLC, long-term l-carnitine treatment; MD, mean diffusivity; NSTLC, no or short-term l-carnitine treatment; RD, radial diffusivity; TBSS, tract-based spatial statistics.

**Figure 4 nutrients-13-01292-f004:**
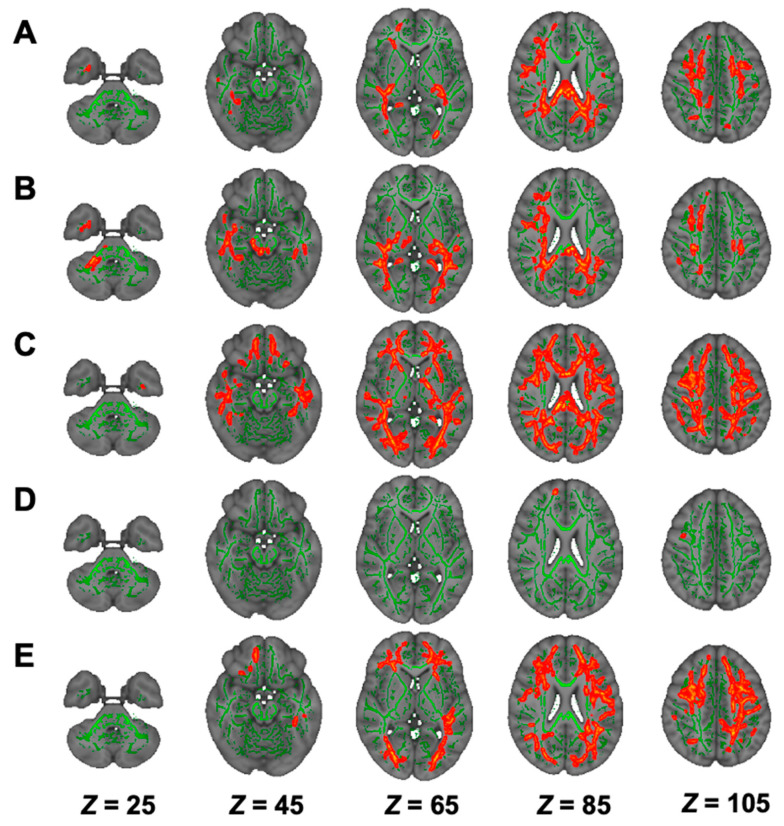
Comparison of TBSS of DKI among hemodialysis patients with NSTLC and LTLC and healthy controls. Comparisons of DKI (AK, (**A**,**B**); RK, (**C**,**D**); and MK, (**E**) indices among hemodialysis patients with NSTLC and LTLC and healthy controls are shown. TBSS analyses show that hemodialysis patients treated with NSTLC have significantly lower AK, RK, and MK than healthy controls (*p* < 0.05, (**A**,**C**,**E**)). Significant but limited low fiber tracts in AK and RK in hemodialysis patients treated with LTLC compared to healthy controls are found (*p* < 0.05, (**B**,**D**)). The skeleton is presented in green. To aid visualization, the results are thickened using the fill script implemented in the FMRIB Software Library. AK, axial kurtosis; DKI, diffusion kurtosis imaging; LTLC, long-term l-carnitine treatment; MK, mean kurtosis; NSTLC, no or short-term l-carnitine treatment; RK, radial kurtosis; TBSS, tract-based spatial statistics.

**Figure 5 nutrients-13-01292-f005:**
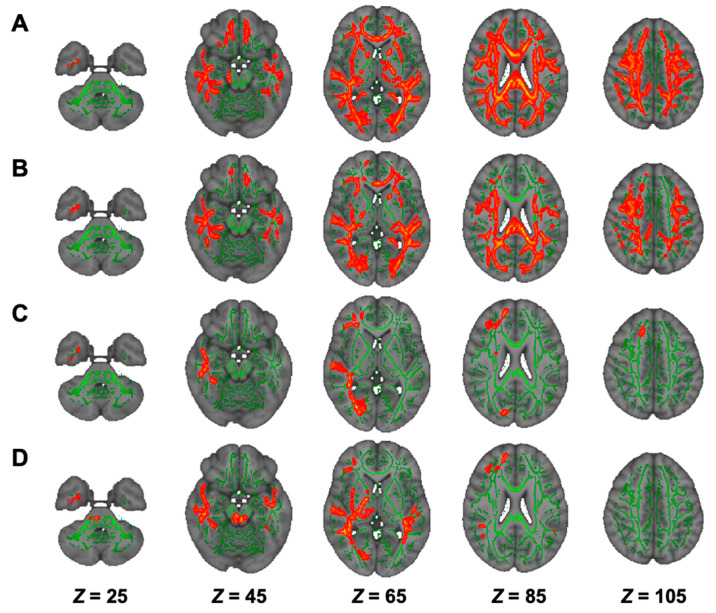
Comparison of TBSS of NODDI among hemodialysis patients treated with NSTLC and LTLC and healthy controls. Comparisons of ICVF (**A**,**B**) and ISO (**C**,**D**) indices among hemodialysis patients treated with NSTLC and LTLC and healthy controls are shown. TBSS analyses show that hemodialysis patients treated with NSTLC (**A**) and LTLC (**B**) have significantly lower ICVF than healthy controls (*p* < 0.05). TBSS analyses show that hemodialysis patients treated with LTLC have significantly lower ISO than hemodialysis patients treated with NSTLC (**C**) and healthy controls (**D**) (*p* < 0.05). The skeleton is presented in green. To aid visualization, the results are thickened using the fill script implemented in the FMRIB Software Library. ICVF, intracellular volume fraction; ISO, isotropic volume fraction; LTLC, long-term l-carnitine treatment; NODDI, neurite orientation dispersion and density imaging; NSTLC, no or short-term l-carnitine treatment; TBSS, tract-based spatial statistics.

**Figure 6 nutrients-13-01292-f006:**
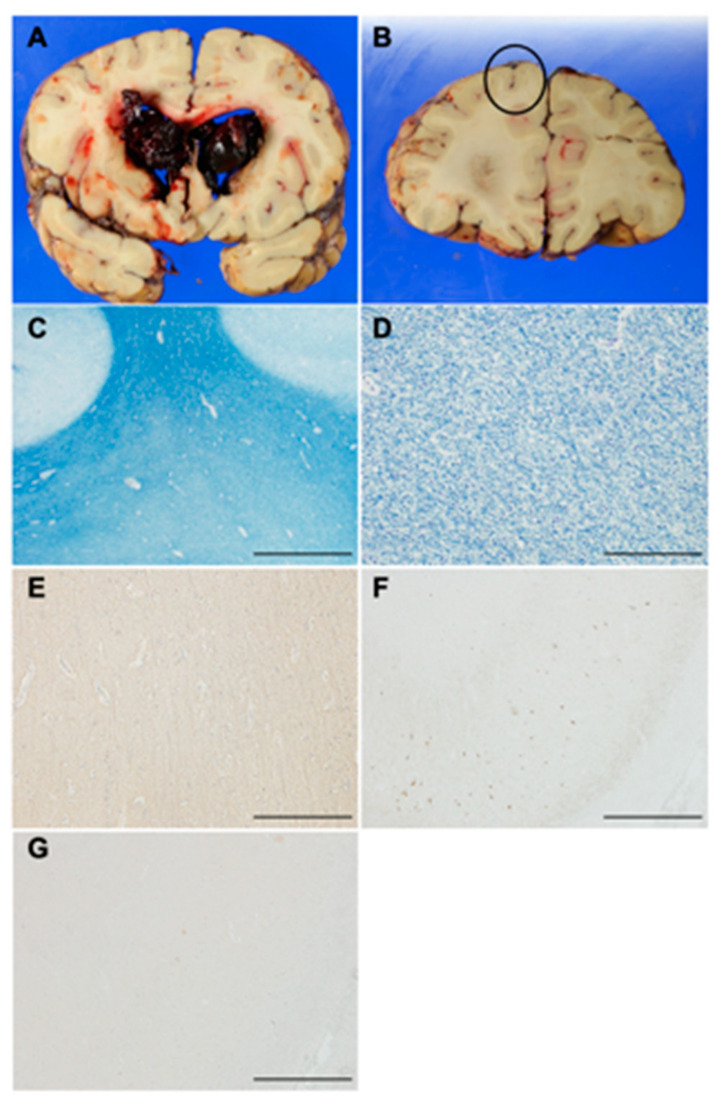
Postmortem study of a case of NSTLC. Macroscopic view of the bilateral cerebral hemispheres shows that right thalamic hemorrhage is comorbid with intraventricular rupture (**A**), but subcortical white mater is preserved (**B**), where histological analysis was carried out ((**B**), circle; (**C**–**E**)). Photomicrographs of Klüber–Barrera ((**C**), scale bar: 5 mm; (**D**), scale bar: 200 μm), neurofilament heavy weight ((**E**), scale bar: 200 μm), phosphorylated tau (**F**), scale bar: 1 mm), and amyloid beta ((**G**), scale bar: 1 mm) staining.

**Table 1 nutrients-13-01292-t001:** Baseline characteristics, laboratory data, and cognitive and radiological findings according to total duration of l-carnitine treatment in hemodialysis patients.

		Duration of LCAR Treatment		*p*
	Total	No or Short-Term	Long-Term	
Characteristic	*n* = 14	*n* = 7	*n* = 7	
Sociodemographic				
Age, years, mean ± SD	70.8 ± 7.2	72.6 ± 9.7	69.0 ± 3.3	0.259
Sex, male, no. (%)	8 (57)	4 (57)	4 (57)	0.589
Final education				0.431
High school	4 (29)	1 (14)	3 (43)	
Business school	2 (14)	1 (14)	1 (14)	
Junior college	3 (21)	1 (14)	2 (29)	
University	5 (36)	4 (57)	1 (14)	
Body mass index	20.6 ± 4.2	21.8 ± 5.3	19.3 ± 2.3	0.456
Risk factors, no. (%)				
Hypertension	13 (93)	7 (100)	6 (86)	1
Diabetes mellitus	6 (43)	3 (43)	3 (43)	0.589
Dyslipidemia	8 (57)	3 (43)	5 (71)	0.589
Current cigarette smoking	1 (7)	1 (14)	0 (0)	1
Coronary artery disease	2 (14)	1 (14)	1 (14)	0.445
Previous history of stroke	3 (21)	2 (29)	1 (14)	1
Cause of renal failure				0.718
Glomerulonephritis	3 (21)	1 (14)	2 (29)	
Diabetic nephropathy	6 (43)	3 (43)	3 (43)	
Nephrosclerosis	1 (7)	1 (14)	0 (0)	
Others	4 (29)	2 (29)	2 (29)	
Total time of hemodialysis	1217 ± 687	870 ± 750	1564 ± 405	0.097
Total amount of LCAR, g	496.4 ± 311.4	241.3 ± 233.1	751.4 ± 62.6	0.002
Laboratory findings, mean ± SD				
Leukocyte count, 102/μL	60.9 ± 16.6	64.1 ± 19.1	57.7 ± 13.4	0.413
LDL-C, mg/dL	84.5 ± 24.1	91.0 ± 28.7	78.0 ± 16.6	0.195
HDL-C, mg/dL	55.5 ± 15.1	51.0 ± 13.9	60.1 ± 15.1	0.082
Triglycerides, mg/dL	109.3 ± 43.7	112.2 ± 43.4	106.3 ± 43.4	0.772
Glucose, mg/dL	133.9 ± 45.5	135.4 ± 44.8	132.4 ± 47.2	0.782
eGFR, mL/min	4.7 ± 2.5	4.3 ± 1.8	5.2 ± 3.1	0.481
Creatinine, mg/dL	10.1 ± 2.6	10.2 ± 3.2	9.9 ± 2.0	0.93
Hs-CRP, mg/dL	0.41 ± 0.65	0.65 ± 0.68	0.17 ± 0.52	<0.001
Calcium, mg/dL	9.0 ± 0.6	8.8 ± 0.8	9.2 ± 0.2	0.173
Phosphate, mg/dL	5.2 ± 1.4	5.5 ± 1.5	5.0 ± 1.3	0.191
Conventional MRI				
PVH, grade 0–3	1.5 ± 0.7	1.4 ± 0.5	1.6 ± 0.8	0.902
DSWMH, grade 0–3	1.1 ± 0.7	1.1 ±0.7	1.0 ±0.8	0.805
Number of CMBs	2.9 ± 6.4	5.1 ± 8.7	0.7 ± 1.0	0.318
Cognitive function test				
MMSE	25.4 ± 3.5	26.0 ± 2.6	24.9 ± 4.3	0.71
HDS-R	26.3 ± 2.3	27.1 ± 1.5	25.4 ± 2.8	0.259
FAB	14.9 ± 3.2	15.0 ± 4.2	14.7 ± 2.0	0.318
MoCA-J	23.9 ± 3.2	23.9 ± 3.8	23.9 ± 2.8	0.71

The chi-squared test and the Mann–Whitney U test were used for comparisons. LCAR = l-carnitine; LDL-C = low-density lipoprotein cholesterol; HDL-C = high-density lipoprotein cholesterol; eGFR = estimated glomerular filtration rate; hs-CRP = high-sensitivity C-reactive protein. MRI = magnetic resonance imaging; PVH = periventricular hyperintensity; DSWMH = deep and subcortical white matter hyperintensity; MMSE = mini mental state examination; HDS-R = Hasegawa dementia rating scale-revised; FAB = frontal assessment battery; MoCA-J = Montreal cognitive assessment Japanese version.

**Table 2 nutrients-13-01292-t002:** Tract-based spatial statistics analysis of diffusion tensor and diffusion kurtosis imaging and neurite orientation dispersion and density imaging in hemodialysis patients and healthy controls.

Modality	Contrast	Cluster Size	Anatomical Region	Peak *t*-Value	Peak MNI Coordinates (*X*, *Y*, *Z*)
DTI					
FA	HC > NSTLC	48,825	Bilateral ATR, corticospinal tract, CCG, forceps minor and major, IFOF, ILF, SLF, SLF temporal part, medial lemniscus, CP, ALIC, PLIC, retrolenticular part of IC, ACR, SCR, PCR, PTR, SS, external capsule, fornix stria terminalis, SFOF, tapetum; left UF, corticospinal tract, ICP, UF; right SCP; MCP, pontine crossing tract, genu, body and splenium of CC, fornix	6.74	(74, 69, 105)
AD	HC < NSTLC	10,699	Bilateral ATR, corticospinal tract, IFOF, SLF, ALIC, PLIC, retrolenticular part of IC, ACR, SCR, PCR, PTR, external capsule, fornix stria terminalis, SFOF; left ILF, SS, tapetum; forceps minor, UF, genu, body and splenium of CC, fornix	5.85	(50, 125, 100)
RD	HC < NSTLC	49,556	Bilateral ATR, corticospinal tract, CCG, IFOF, ILF, SLF, UF, SLF temporal part, CP, ALIC, PLIC, retrolenticular part of IC, ACR, SCR, PCR, PTR, SS, external capsule, fornix stria terminalis, SFOF, tapetum; forceps minor and major, genu, body and splenium of CC, fornix	7.43	(140, 117, 50)
MD	HC < NSTLC	43,676	Bilateral ATR, corticospinal tract, CCG, IFOF, ILF, SLF, UF, SLF temporal part, ALIC, PLIC, retrolenticular part of IC, ACR, SCR, PCR, PTR, SS, external capsule, fornix stria terminalis, SFOF, tapetum; left CHp; forceps minor and major, genu, body and splenium of CC, fornix	7.10	(53, 104, 105)
DKI					
AK	HC > NSTLC	15,653	Bilateral corticospinal tract, IFOF, ILF, SLF, PLIC, retrolenticular part of IC, ACR, SCR, PCR, PTR, external capsule, fornix stria terminalis, tapetum; right ATR, cingulum hippocampus, UF, SS, CCG, CHp; forceps minor and major, body and splenium of CC	6.03	(53, 65, 61)
	HC > LTLC	16,138	Bilateral corticospinal tract, CHp, IFOF, ILF, SLF, SLF temporal part, corticospinal tract, medial lemniscus, SCP, CP, PLIC, retrolenticular part of IC, SCR, PCR, PTR, SS, fornix stria terminalis, tapetum; left ATR, UF, ALIC, ACR, external capsule, SFOF; forceps major, MCP, pontine crossing tract, body and splenium of CC	6.19	(55, 92, 82)
RK	HC > NSTLC	45,239	Bilateral ATR, corticospinal tract, CCG, IFOF, ILF, SLF, UF, SLF temporal part, ALIC, retrolenticular part of IC, ACR, SCR, PCR, PTR, SS, external capsule, fornix stria terminalis, SFOF; right UF, tapetum; forceps minor and major, genu, body and splenium of CC	6.11	(98, 172, 110)
	HC > LTLC	69	forceps minor	4.89	(79, 177, 101)
MK	HC > NSTLC	28,546	Bilateral ATR, IFOF, ILF, SLF, ALIC, ACR, SCR, PCR, PTR, SS, external capsule, SFOF; left corticospinal tract, UF, retrolenticular part of IC; forceps minor and major, genu, body and splenium of CC	6.82	(97, 141, 131)
NODDI					
ICVF	HC > NSTLC	67,959	Bilateral ATR, corticospinal tract, CCG, IFOF, ILF, SLF, UF, SLF temporal part, CP, ALIC, PLIC, retrolenticular part of IC, ACR, SCR, PCR, PTR, SS, external capsule, CHp, fornix stria terminalis, SFOF, tapetum; left UF; right CHp; forceps minor and major, genu, body and splenium of CC	8.15	(106, 82, 84)
	HC > LTLC	38,021	Bilateral ATR, corticospinal tract, CCG, IFOF, ILF, SLF, UF, SLF temporal part, ALIC, PLIC, retrolenticular part of IC, ACR, SCR, PCR, PTR, SS, external capsule, CHp, fornix stria terminalis, SFOF; left tapetum; right CHp, CP; forceps minor and major, genu, body and splenium of CC	6.22	(113, 63, 103)
ISO	HC > LTLC	3978	right ATR, Inferior fronto-occipital fasciculus, ILF, UF, retrolenticular part of IC, ACR, SCR, PTR, SS; forceps minor and major	5.88	(48, 116, 49)
	NSTLC > LTLC	8312	Bilateral corticospinal tract, IFOF, ILF, SLF, UF, SCP, CP, retrolenticular part of IC, PTR, SS; right ATR, CHp, SLF temporal part, ALIC, PLIC, external capsule, fornix stria terminalis; forceps minor and major, MCP, pontine crossing tract	7.52	(129, 83, 68)

DTI = diffusion tensor imaging; DKI = diffusion kurtosis imaging; NODDI = neurite orientation dispersion and density imaging; ICVF = intracellular volume fraction; ODI = orientation dispersion index; ISO = isotropic volume fraction; HC = healthy control; NSTLC = no or short-term l-carnitine treatment; LTLC = long-term l-carnitine treatment; ATR = anterior thalamic radiation; CCG = cingulum in cingulate gyrus; IFOF = inferior fronto-occipital fasciculus; ILF = inferior longitudinal fasciculus; SLF = superior longitudinal fasciculus; CP = cerebral peduncle; ALIC = anterior limb of internal capsule; PLIC = posterior limb of internal capsule; ACR = anterior corona radiata; SCR = superior corona radiata; PCR = posterior corona radiata; PTR = posterior thalamic radiation; SS = striatum; SFOF = superior frontal occipital fasciculus; UF = uncinate fasciculus; ICP = inferior cerebellar peduncle; SCP = superior cerebellar peduncle; MCP = middle cerebellar peduncle; CC = corpus callosum.

**Table 3 nutrients-13-01292-t003:** Association of completion time of Trail-making test A with tracts-of-interest on DTI, DKI, and NODDI between NSTLC and LTLC groups.

Region	FA		AD		RD		MD		AK		RK		MK		ICVF		ISO	
	*R*	*p*	*R*	*p*	*R*	*p*	*R*	*p*	*R*	*p*	*R*	*p*	*R*	*p*	*R*	*p*	*R*	*p*
Whole			0.628	0.016			0.604	0.022	−0.741	0.002 *	−0.591	0.026	−0.609	0.021	−0.547	0.043		
Lcc	−0.543	0.045							−0.745	0.002 *								
Rcc	−0.609	0.021			0.669	0.009	0.591	0.026			−0.565	0.035						
Latr			0.679	0.008			0.6	0.023	−0.771	0.001 *					−0.613	0.02		
Ratr			0.793	0.001 *	0.554	0.04	0.648	0.012	−0.248	0.392							0.609	0.021
Lcs			0.644	0.013														
Rcs			0.569	0.034			0.559	0.038										
fm					0.569	0.034	0.678	0.008							−0.789	0.001 *		
fmi									−0.574	0.032			−0.644	0.013	−0.534	0.049		
Lifof									−0.613	0.02					−0.582	0.029		
Rifof											−0.648	0.012	−0.591	0.026				
Lslf									−0.613	0.02								
Rslf									−0.758	0.002 *					−0.574	0.032		
Lilf									−0.556	0.039								
Rilf											−0.578	0.03						
Rslftemp											−0.587	0.027						

Spearman’s rank correlation coefficient was used for comparison. * Values differed significantly (*p* < 0.05, FDR corrected). LCAR = l-carnitine; LTLC = large amount of total l-carnitine treatment; NSTLC = no or short amount of total l-carnitine treatment; DTI = diffusion tensor imaging; DKI = diffusion kurtosis imaging; NODDI = neurite orientation dispersion and density imaging; FA = fractional anisotropy; AD = axial diffusivity; RD = radial diffusivity; MD = mean diffusivity; AK = axial kurtosis; RK = radial kurtosis; MK = mean kurtosis; ICVF = intra-cellular volume fraction; ISO = isotropic volume fraction; ODI = orientation dispersion index; FDR = false detection rate; L(R)atr = left(right) anterior thalamic radiation; L(R)cs = corticospinal tract; L(R)cc = cingulum (cingulate gyrus); L(R)ch = cingulum (hippocampus); fm = forceps major; fmi = forceps minor; L(R)ifof = left(right) inferior fronto-occipital fasciculus; L(R)ilf = left(right) inferior longitudinal fasciculus; L(R)slf = left(right) superior longitudinal fasciculus; L(R)uf = left(right) uncinate fasciculus; L(R)slftemp = left(right) superior longitudinal fasciculus temporal part.

## Data Availability

The datasets used and analyzed during the current study are available from the corresponding author upon reasonable request.

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
