# Peer review of "Possible Neuroprotective Effects of l-Carnitine on White-Matter Microstructural Damage and Cognitive Decline in Hemodialysis Patients"

_nutrients, 2021, doi:10.3390/nu13041292_

Round 1
Reviewer 1 Report
The study entitled “Neuroprotective effects of L-carnitine on vascular dementia in hemodialysis patients” by Ueno and collaborators aims to demonstrate the beneficial effects of the L-carnitine on the white matter tracts damage and cognitive dysfunction impairments suffered by hemodialysis patients.
Although the study is interesting, it has some methodological concerns that limit it, beyond the honest considerations described by the own authors in the section “limitations of the study”.
Particularly, it is quite confusing the recruitment of the hemodialysis patients and healthy controls described in the “study participants”. In this same section, the way to indicate the sex of the participants per group (mentioning only the men) sounds at least strange.
In relation to the section “drug administration and classification“, it is also unclear the dose schedule as well as the duration of the treatment received by each group.
This general lack of precision in the “study design” seems to be consequence of some of the limitations of the study (really explained by the authors), but the scarce number of participants it is not the only concern. Another one is the difficulty to identify retrospectively the duration of L-carnitine treatment; fact, also reflected in the methodology.
Additionally, the fact that only have a single patient to justified the histopathological analyses adds new limitations to the study that are even more increased by the diversity of causes that led patients to be hemodialyzed. Moreover, the multiple medications received by the enrolled patients could also influence differentially the goals of the study, as indicated by the authors themselves.
In conclusion, although the study is interesting and the MRI-techniques, neuropsychological tests and statistical analysis are well conducted, the main objective pursued about the benefits of the long-term treatment with L-carnitine on the white matter microstructural damage and cognitive impairment in hemodialysis patients is quite limited. Thus, the objectives of the study should be redefined and reflected in the summary and conclusions; finally, the title would also have to be modified accordingly
Author Response
We appreciate the positive evaluations of our manuscript by the reviewers. The following text provides our point-by-point responses to each comment from the reviewers. Line numbers mentioned in the response to each comment refer to the numbers appearing in the left margin of the revised manuscript.
Reviewer 1
The study entitled “Neuroprotective effects of L-carnitine on vascular dementia in hemodialysis patients” by Ueno and collaborators aims to demonstrate the beneficial effects of the L-carnitine on the white matter tracts damage and cognitive dysfunction impairments suffered by hemodialysis patients.
Although the study is interesting, it has some methodological concerns that limit it, beyond the honest considerations described by the own authors in the section “limitations of the study”.
Reply: We thank the reviewer for this important comment. As suggested by the reviewer, the present study has methodological concerns. In particular, retrospective investigation of the profiles of l-carnitine therapy is a major limitation of the study. We revised our manuscript and provide our point-by-point responses for the following comments.
Particularly, it is quite confusing the recruitment of the hemodialysis patients and healthy controls described in the “study participants”. In this same section, the way to indicate the sex of the participants per group (mentioning only the men) sounds at least strange.
Reply: Thank you very much for this comment. As indicated by the reviewer, the section on “study participants” was slightly misleading. We stated the inclusion and exclusion criteria first, and separated the statements related to study subjects and controls. We also added the exact number of women and men among the study subjects and controls.
Line 19, page 1: Fourteen hemodialysis patients underwent dMRI and laboratory and neuropsychological tests, which were compared between seven patients each in two groups based on duration of l-carnitine treatment: (1) no or short-term l-carnitine treatment (NSTLC); and (2) long-term l-carnitine treatment (LTLC). Ten age and sex-matched controls were enrolled.
Line 83, page 2: Patients were included based on the following inclusion criteria: (i) age between 55 and 90 years; (ii) undergoing hemodialysis in Juntendo University Hospital Dialysis Center; and (iii) having at least one atherosclerotic vascular risk factor. Exclusion criteria were: (i) any contraindication to brain MRI; (ii) moderate to severe cognitive decline or aphasia (need some assistance to complete their daily living activities or communication); (iii) coexistence of neurological diseases; and (iv) presence of severe pneumonia, chronic heart failure, or advanced cancer. Previous studies with sample sizes of 10 to 17 participants have been conducted to explore the pathophysiology of microstructural alterations of white matter tracts using NODDI [17,18]. Thus, 14 maintenance hemodialysis patients (age 70.8±7.2 years, 6 women and 8 men), treated three times a week at outpatient hemodialysis units in Juntendo University Hospital, were included in the present study. In addition, 10 age- and sex-matched healthy controls (73.0±4.4 years, 5 women and 5 men) who were literate or able to be interviewed and could communicate effectively, with no history of severe mental disorders or dementia, were also recruited. This study was conducted in accordance with the Declaration of Helsinki. The independent ethics committee of Juntendo University Hospital approved this study (16-170). All study subjects were given an explanation of the study, and written, informed consent was obtained for the study objective, MRI and cognitive function tests, enrolment in the study, and ensuring confidentiality of information.
In relation to the section “drug administration and classification“, it is also unclear the dose schedule as well as the duration of the treatment received by each group.
Reply: We thank the reviewer for this comment. As suggested, the dose schedule and duration of l-carnitine treatment in the NSTLC and LTLC groups were unclear. We have included the detailed protocol of l-carnitine treatment in the ‘Drug administration and classification’ section.
Line 112, page 3: The protocol for treatment with l-carnitine (Otsuka Pharmaceutical Co., Ltd., Tokushima, Japan) was 600 mg orally once a day starting in July 2012, followed by 1000 mg intravenously per hemodialysis day starting in January 2014. Termination of l-carnitine treatment was determined randomly, and patients without l-carnitine treatment were also included. Based on the duration of l-carnitine treatment, 7 patients each were classified into the no or short-term l-carnitine treatment (NSTLC) and LTLC long-term l-carnitine treatment (LTLC) groups, with an allocation ratio of 1:1. In the NSTLC group, 4 patients’ intravenous l-carnitine treatment was stopped from June 2014 to April 2015, and thus l-carnitine was discontinued on the day of dMRI and neuropsychological tests (January 2017 to September 2017), whereas the remaining 3 patients were not treated with l-carnitine. In the LTLC group, intravenous l-carnitine treatment was continued on the day of dMRI and neuropsychological tests (January 2017 to September 2017) in 7 patients.
This general lack of precision in the “study design” seems to be consequence of some of the limitations of the study (really explained by the authors), but the scarce number of participants it is not the only concern. Another one is the difficulty to identify retrospectively the duration of L-carnitine treatment; fact, also reflected in the methodology.
Reply: We thank the reviewer for this comment. We agree with the reviewer that our study has not only a small sample size, but also methodological issues, especially as a cross-sectional study and a retrospective investigation of the profiles of l-carnitine therapy. We have stated these concerns in the study limitations in greater detail. Furthermore, we revised the title, abstract, and conclusions describing the therapeutically limited effects of l-carnitine.
Line 501, page 16: First, the sample size was small, and this study was a non-randomized, cross-sectional study analyzing the therapeutic effects of l-carnitine for cognitive decline and microstructural damage on dMRI in hemodialysis patients. Although there was a significant difference in the total amount of l-carnitine between the LTLC and NSTLC groups, the protocol of l-carnitine treatment was quite complicated, and the NSTLC group was heterogenous in that it consisted of patients without l-carnitine treatment and patients with short-term l-carnitine treatment. Thus, to explore the protective effects of l-carnitine for disorders of white matter fiber tracts and cognitive function, large-scale, prospective longitudinal studies with an appropriate trial design are warranted.
Line 2, page 1: Possible neuroprotective effects of l-carnitine on white matter microstructural damage and cognitive decline in hemodialysis patients
Line 29, page 1: Long-term treatment with l-carnitine might alleviate white matter microstructural damage and cognitive impairment in hemodialysis patients.
Line 520, page 17: The current data showed that l-carnitine may have neuroprotective roles against the white matter microstructural damage and cognitive impairment in hemodialysis patients.
Additionally, the fact that only have a single patient to justified the histopathological analyses adds new limitations to the study that are even more increased by the diversity of causes that led patients to be hemodialyzed. Moreover, the multiple medications received by the enrolled patients could also influence differentially the goals of the study, as indicated by the authors themselves.
Reply: We thank the reviewer for this comment. As suggested by the reviewer, the current study has additional limitations. First, only one postmortem study was carried out. Since the causes of renal failure were diverse, there is likely to have been pathological heterogeneity. Second, hemodialysis patients not infrequently have multiple atherosclerotic risk factors, as well as cardiovascular disease and stroke. Therapeutic agents for these comorbidities, such as statins, antiplatelet drugs, and angiotensin II receptor blockers, might have antiatherogenic roles and modify the alterations of the microstructure of white matter tracts and the neuropsychological status in hemodialysis patients.
Line 511, page 17: Second, only one postmortem examination was performed in the current study. The types of end-stage renal diseases in the enrolled patients varied, and therefore there may have been heterogeneity in postmortem brain pathology.
Line 514, page 17: Third, other medications including statins, antiplatelet drugs, and angiotensin II receptor blockers might have antiatherogenic effects and could have affected the microstructural damage of white matter tracts and the neuropsychological status in hemodialysis patients.
In conclusion, although the study is interesting and the MRI-techniques, neuropsychological tests and statistical analysis are well conducted, the main objective pursued about the benefits of the long-term treatment with L-carnitine on the white matter microstructural damage and cognitive impairment in hemodialysis patients is quite limited. Thus, the objectives of the study should be redefined and reflected in the summary and conclusions; finally, the title would also have to be modified accordingly.
Reply: We thank the reviewer for this important comment. As suggested by the reviewer, the essential aim of the current study was to elucidate the treatment effects of l-carnitine on microstructural damage by dMRI and cognitive decline in hemodialysis patients. We stated this in the revised abstract and conclusions. We also revised the title to ‘Possible neuroprotective effects of l-carnitine on white matter microstructural damage and cognitive decline in hemodialysis patients’.
Line 2, page 1: Possible neuroprotective effects of l-carnitine on white matter microstructural damage and cognitive decline in hemodialysis patients
Line 15, page 1: Although l-carnitine alleviated white matter lesions in an experimental study, the treatment effects of l-carnitine on white matter microstructural damage and cognitive decline in hemodialysis patients are unknown.
Line 27, page 1: The LTLC group showed better achievement on the Trail Making Test-A, which was correlated with amelioration of disorders in some white matter tracts. Novel dMRI tractography detected abnormalities of white matter tracts after hemodialysis.
Line 75, page 2: L-carnitine alleviated white matter lesions and cognitive impairment in an experimental study, but its effects on cognitive decline in hemodialysis patients are essentially unknown. The aim of the current study was to elucidate the therapeutic efficacy of l-carnitine for disorders of white matter tracts seen on dMRI and cognitive decline in hemodialysis patients.
Line 520, page 17: Current data indicated that l-carnitine may have neuroprotective roles for the white matter microstructural damage and cognitive impairment in hemodialysis patients.
Reviewer 2 Report
The authors explored the disorder of white matter tracts using dMRI and the impact of L-carnitine treatment for the white matter tract injuries and cognitive dysfunction, in hemodialysis patients. They stated that the impairment of white matter tracts was recorded in hemodialysis patients as well as these injuries were alleviated by long-term L-carnitine treatment, along with reduction of hs-CRP levels. Moreover, they discussed that hemodialysis patients treated with long-term L-carnitine treatment displayed better performance on TMT-A than hemodialysis patients with no or short-term L-carnitine treatment.
Altogether the results obtained with L-carnitine are interesting, the experimental design show some criticisms:
- The sample size was very small. They should justify the experimental design with a small number of subjects in aims of the introduction section. Another important aspect to justify is the choice to make an experimental group “no or short-term L-carnitine treatment” where they combine treated and untreated subjects.
- The authors compared the “no or short-term L-carnitine treatment group” vs “long-term L-carnitine treatment group”. In particular, they used 600 mg and 1000 mg of creatinine by oral and intravenous routes, respectively, making a critical comparison. It is note that the oral route can affect the half-life of creatinine as well as its metabolite profile. However, they evaluated total, free and acylated creatinine in both treatment groups. In this regard, they should perform the statistical analysis of correlation between the creatinine plasmatic levels (free and acylated creatinine) and the white matter microstructural changes as well as cognitive decline. This knowledge can help to evaluate and discuss the real contribution of neuroprotection by free creatinine in these subjects.
Author Response
We appreciate the positive evaluations of our manuscript by the reviewers. The following text provides our point-by-point responses to each comment from the reviewers. Line numbers mentioned in the response to each comment refer to the numbers appearing in the left margin of the revised manuscript.
Reviewer 2
The authors explored the disorder of white matter tracts using dMRI and the impact of L-carnitine treatment for the white matter tract injuries and cognitive dysfunction, in hemodialysis patients. They stated that the impairment of white matter tracts was recorded in hemodialysis patients as well as these injuries were alleviated by long-term L-carnitine treatment, along with reduction of hs-CRP levels. Moreover, they discussed that hemodialysis patients treated with long-term L-carnitine treatment displayed better performance on TMT-A than hemodialysis patients with no or short-term L-carnitine treatment.
Altogether the results obtained with L-carnitine are interesting, the experimental design show some criticisms:
Reply: We thank the reviewer for this comment. As suggested, the study has not only an issue regarding small sample size, but also other potential concerns, such as being a cross-sectional, retrospective investigation of profiles of l-carnitine therapy, including only one autopsy case study, and the impact of other therapeutic agents on white matter microstructural damage. We described the study limitations in greater detail (line 500, page 16), and we revised the title, abstract, and conclusions stating the therapeutically limited effects of l-carnitine.
Line 2, page 1: Possible neuroprotective effects of l-carnitine on white matter microstructural damage and cognitive decline in hemodialysis patients
Line 29, page 1: Long-term treatment with l-carnitine might alleviate white matter microstructural damage and cognitive impairment in hemodialysis patients.
Line 520, page 17: The current data showed that l-carnitine may have neuroprotective roles against the white matter microstructural damage and cognitive impairment in hemodialysis patients.
The sample size was very small. They should justify the experimental design with a small number of subjects in aims of the introduction section. Another important aspect to justify is the choice to make an experimental group “no or short-term L-carnitine treatment” where they combine treated and untreated subjects.
Reply: We thank the reviewer for this comment. Previous studies with small sample sizes of 10 to 17 participants have been conducted to explore the pathophysiology of microstructure damage of white matter tracts using NODDI [17, 18]. Thus, we decided to enroll 14 patients in the current study. We referred to those studies and stated the justification for the sample size in the Introduction and Methods. As for “no or short-term l-carnitine treatment”, we retrospectively investigated the treatment profile of l-carnitine in 14 hemodialysis patients, including 3 patients without treatment, and 7 patients each were classified into the no or short-term l-carnitine treatment (NSTLC) and LTLC long-term l-carnitine treatment (LTLC) groups with an allocation ratio of 1:1. Although there was a significant difference in the total amount of l-carnitine between the LTLC and NSTLC groups, the protocol of l-carnitine treatment was quite complicated, which was a major issue in our study. Thus, to explore the protective effects of l-carnitine for disorders of white matter fiber tracts and cognitive function, large-scale, prospective, longitudinal studies with an appropriate trial design are warranted. We described the detailed treatment protocol of l-carnitine in the ‘Drug administration and classification’ section, and the limitations of the study.
Line 73, page 2: Meanwhile, several studies with small sample sizes of 10 to 17 participants have been conducted to explore the pathophysiology of microstructural damage of white matter tracts using NODDI [17,18].
Line 88, page 2: Previous studies with sample sizes of 10 to 17 participants have been conducted to explore the pathophysiology of microstructural alterations of white matter tracts using NODDI [17,18]. Thus, 14 maintenance hemodialysis patients (age 70.8±7.2 years, 6 women and 8 men), treated three times a week at outpatient hemodialysis units in Juntendo University Hospital, were included in the present study.
Line 112, page 3: The protocol for treatment with l-carnitine (Otsuka Pharmaceutical Co., Ltd., Tokushima, Japan) was 600 mg orally once a day starting in July 2012, followed by 1000 mg intravenously per hemodialysis day starting in January 2014. Termination of l-carnitine treatment was determined randomly, and patients without l-carnitine treatment were also included. Based on the duration of l-carnitine treatment, 7 patients each were classified into the no or short-term l-carnitine treatment (NSTLC) and LTLC long-term l-carnitine treatment (LTLC) groups, with an allocation ratio of 1:1. In the NSTLC group, 4 patients’ intravenous l-carnitine treatment was stopped from June 2014 to April 2015, and thus l-carnitine was discontinued on the day of dMRI and neuropsychological tests (January 2017 to September 2017), whereas the remaining 3 patients were not treated with l-carnitine. In the LTLC group, intravenous l-carnitine treatment was continued on the day of dMRI and neuropsychological tests (January 2017 to September 2017) in 7 patients.
Line 501, page 16: First, the sample size was small, and this study was a non-randomized, cross-sectional study analyzing the therapeutic effects of l-carnitine for cognitive decline and microstructural damage on dMRI in hemodialysis patients. Although there was a significant difference in the total amount of l-carnitine between the LTLC and NSTLC groups, the protocol of l-carnitine treatment was quite complicated, and the NSTLC group was heterogenous in that it consisted of patients without l-carnitine treatment and patients with short-term l-carnitine treatment. Thus, to explore the protective effects of l-carnitine for disorders of white matter fiber tracts and cognitive function, large-scale, prospective longitudinal studies with an appropriate trial design are warranted.
Reference
17.Mastropietro, A.; Rizzo, G.; Fontana, L.; Figini, M.; Bernardini, B.; Straffi, L.; Marcheselli, S.; Ghirmai, S.; Nuzzi, N.P.; Malosio, M.L., et al. Microstructural characterization of corticospinal tract in subacute and chronic stroke patients with distal lesions by means of advanced diffusion MRI. Neuroradiology 2019, 61, 1033-1045, doi:10.1007/s00234-019-02249-2.
18.Nemanich, S.T.; Mueller, B.A.; Gillick, B.T. Neurite orientation dispersion and density imaging quantifies corticospinal tract microstructural organization in children with unilateral cerebral palsy. Hum Brain Mapp 2019, 40, 4888-4900, doi:10.1002/hbm.24744.
The authors compared the “no or short-term L-carnitine treatment group” vs “long-term L-carnitine treatment group”. In particular, they used 600 mg and 1000 mg of creatinine by oral and intravenous routes, respectively, making a critical comparison. It is note that the oral route can affect the half-life of creatinine as well as its metabolite profile. However, they evaluated total, free and acylated creatinine in both treatment groups. In this regard, they should perform the statistical analysis of correlation between the creatinine plasmatic levels (free and acylated creatinine) and the white matter microstructural changes as well as cognitive decline. This knowledge can help to evaluate and discuss the real contribution of neuroprotection by free creatinine in these subjects.
Reply: We agree with the reviewer’s opinion that it is critical to analyze the association of serum total, free, and acylated carnitine concentrations with white matter microstructural changes on diffusion MRI. This would be critical to explore the direct effect of free creatinine for neuroprotection. However, the current study has methodological issues with a complicated protocol for l-carnitine treatment. In particular, the no or short-term treatment with l-carnitine group included 4 patients with short-term treatment with l-carnitine, in whom carnitine concentrations were higher, but treatment was discontinued, and MRI and neuropsychological tests were then carried out. Regrettably, it is impossible to investigate the association of carnitine concentrations with dMRI data in the current study. We described the treatment protocol for l-carnitine in the ‘Drug administration and classification’ section in greater detail (Line 111, page 3), and described this important study direction in the limitations.
Line 509, page 16: Furthermore, it is important to explore the association of serum carnitine concentrations with microstructural damage of cerebral white matter.
Round 2
Reviewer 1 Report
The study is interesting and have been improved, particularly in relation the the clarification of the methodological concerns. However, I encourage the authors to design a new trial to solve the limitations of the study.
Reviewer 2 Report
The authors replied appropriately to the questions and improved the manuscript